# Modelling Spirals of Silence and Echo Chambers by Learning from the Feedback of Others

**DOI:** 10.3390/e24101484

**Published:** 2022-10-18

**Authors:** Sven Banisch, Felix Gaisbauer, Eckehard Olbrich

**Affiliations:** 1Institute of Technology Futures, Karlsruhe Institute of Technology, 76133 Karlsruhe, Germany; 2Max Planck Institute for Mathematics in the Sciences, 04103 Leipzig, Germany

**Keywords:** social dynamics, group dynamics, spiral of silence, echo chambers, silent majorities, reinforcement learning, social feedback, social neuroscience, opinion dynamics, 91D30, 91F10, 00A69

## Abstract

What are the mechanisms by which groups with certain opinions gain public voice and force others holding a different view into silence? Furthermore, how does social media play into this? Drawing on neuroscientific insights into the processing of social feedback, we develop a theoretical model that allows us to address these questions. In repeated interactions, individuals learn whether their opinion meets public approval and refrain from expressing their standpoint if it is socially sanctioned. In a social network sorted around opinions, an agent forms a distorted impression of public opinion enforced by the communicative activity of the different camps. Even strong majorities can be forced into silence if a minority acts as a cohesive whole. On the other hand, the strong social organisation around opinions enabled by digital platforms favours collective regimes in which opposing voices are expressed and compete for primacy in public. This paper highlights the role that the basic mechanisms of social information processing play in massive computer-mediated interactions on opinions.

## 1. Introduction

A better understanding of the collective processes underlying public opinion expression is crucial for a better understanding of modern society. Sociological models drawing on network science [1,2,3] and basic principles of human interaction behaviour [4,5] have already provided useful insights into collective phenomena related to mass mobilisation [6,7,8], societal-level change of behaviour [9,10,11] and beliefs [12]. However, for a change to happen and for a movement to gain pace, the alternative must be voiced by a sufficiently large group [13]. Furthermore, to be voiced, it must be perceived as something that can be said without “fear of isolation” [14].

The spiral of silence theory [15] is based on the old “law of opinion” ([14] John Locke). It focuses on the collective perception of what can be publicly voiced and hence impact the further perception of public opinion. The theory assumes that humans possess a “quasi-statistical organ” [16] to perceive what can be said without being socially sanctioned and explains public opinion dynamics as a spiralling process in which silence may lead to more silence. In this paper, we propose a mathematical model for this process based on reinforcement learning (RL) by social feedback [17]. In repeated games played over a network, agents receive signals of approval or disapproval for expressing their opinion to peers. Agents evolve an expectation about the social reward obtained when expressing their opinion and remain silent if they expect punishment (negative reward). In this way, social feedback dynamics naturally capture the assumed “quasi-statistical” perception of the opinion landscape surrounding an agent.

Our paper develops a computational model that captures the basic assumptions of the spiral of silence theory and grounds it in neuroscientific research on social information processing. While the spiral of silence theory frequently refers to the “social nature of man” [18], no attempt of grounding this assumption in social psychology and neuroscience has been made. On the other hand, the potential for explaining collective behaviour based on mechanisms identified in cognitive and social neuroscience is frequently emphasized [19,20,21], but its integration with sociological theories of collective opinion expression [6,7,15] is lacking. The social feedback theory (SFT) bridges this gap by formulating collective processes of opinion expression as a multiagent problem in which individual agents adapt according to a reward- and value-based learning scheme identified in neuroscientific research [22,23,24,25,26,27]. With these repeated opinion expression games, the SFT provides a coherent framework for modelling collective opinion processes that integrates basic neuroscientific findings, adaptive decision-making [28] and a political theory of public opinion [15,16].

Given that the processing of and learning by social feedback is so deeply rooted in the human brain, it is of the uttermost importance to better understand the collective consequences of these processes. Especially in social media environments, a tremendous number of quick feedback decisions is made day by day by billions of users. “Like buttons” and quantitative markers of collective endorsement can be associated with low cognitive costs, which suggests that a dominant role is played by the fast value-processing mechanisms accounted for by RL. Recent studies have provided evidence for that [29,30]. While it is reasonable to assume that the social reward circuit has evolved to facilitate cohesion and cooperation in small groups [20] with intensive pair-bonding [31], this reasoning may not apply for societies of increased complexity [32,33]. In complex social networks, the human ability to coordinate with in-groups may come at the expense of an increasing alienation to out-groups and therefore drive polarization dynamics [17]. Here, we show that social feedback dynamics provide a neurobiologically grounded explanation of collective processes involved in “spirals of silence” [15] and analyse how structural transformations enabled by social media give voice to groups that previously went unheard.

Providing a mechanism-based approach [5] to model the phenomena of collective opinion expression or silence enables a more general application of the assumptions underlying the spiral of silence theory. Most importantly, our model allows us to relate structural variations across different opinion groups to different regimes of collective opinion expression. In this regard, we show that social feedback mechanisms may lead to spirals of silence in unstructured random networks, but that the same mechanism generates highly active echo chambers if social networks become more assortative and homophilous with respect to opinions.

## 2. Model

### 2.1. Social Feedback Processing in the Brain

Our modelling choices are well-grounded in neuroscientific insights into human social nature. Social neuroscience aims to identify neural mechanisms involved into the processing of social cues. fMRI studies have shed light on the interaction and interconnectedness of different brain regions and their functional role in social cognition. While it has long been controversial whether human nature evolved a neural circuity specifically for handling social information or not [20,24,31,34], it is now relatively settled that a basic “reinforcement circuit” [23,35] is strongly involved into value-based decisions and learning from social feedback [29,30,36,37,38,39]. Other brain processes interfere with this circuity [20,23,40], especially when social situations and tasks involve higher cognitive functions such as trust [35], morality [41] or representations of self and the other [42,43].

Temporal difference reinforcement learning (TDRL) [44,45] has provided a useful computational account of the brain mechanisms underlying social reward processing and learning [24,25,26,27]. In TDRL, a new estimate of the expected value $Q^{t+1}$ associated with an action is a function of the current estimate $Q^{t}$ and the temporal difference (TD) error $\delta^{t}$ between this estimate and the reward that is actually obtained: $Q^{t+1}=Q^{t}+\alpha \delta^{t}$. With a rate governed by α (referred to as learning rate), this scheme converges to a stable equilibrium in which the TD error $\delta^{t}$ approaches zero such that the expectations and actual reception of rewards are aligned [45]. The usefulness of TDRL in computational neuroscience derives from the finding that the activity of dopaminergic neurons in the midbrain regions is quantitatively related to the “reward–prediction error” [20] between the experienced reward and its expected value [22,46,47], that is, to $\delta_{t}$. Social neuroscience has provided ample evidence that such a basic reward processing circuit is also highly involved into peer influence processes [29,38], social conformity [37] and approval [39].

### 2.2. Opinion Expression Games

We consider the situation that two groups with different standpoints on a controversial issue have evolved and engage in public discourse. In contrast to most existing opinion dynamics models, we consider that the opinions of agents are fixed, because we want to understand the conditions under which agents with a given opinion become silent. Individuals within both opinion groups have two available actions: they can decide to express (E) their standpoint or to be silent (S). They receive supportive feedback from their respective in-group and negative feedback from agents in the out-group when expressing their opinion. Individual interaction is hence formulated as repeated opinion expression games with a reward system that captures approval and disapproval by peers:(1)$$r_{i}^{t}=\left\{\begin{array}{cc} -c & \mathrm{silent}\;\mathrm{neighbour}\\ -c+1 & \;\mathrm{agreement}\\ -c-1 & \;\mathrm{disagreement} \end{array}\right.$$
The parameter *c* corresponds to a fixed cost of opinion expression and *i* refers to the individual agent. Having received a social reward during an interaction, agents update the expected value $Q_{i}\left(A\right)$ of their current action by TDRL
(2)$$Q_{i}{\left(A\right)}^{t+1}=Q_{i}{\left(A\right)}^{t}+\alpha \underset{\mathrm{TD}\;\mathrm{error}}{\underset{︸}{\left(r_{i}^{t}-Q_{i}{\left(A\right)}^{t}\right)}}$$
with learning rate α. As the reward of silence (S) is zero in the game, we only have to keep track of the value for the opinion expression and skip action indices in the sequel ($Q_{i}\left(E\right)=Q_{i}$). $Q_{i}$ hence accounts for the subjective reward that agent *i* expects when expressing their opinion, and the agent will remain silent if this value is negative. Given the current value of opinion expression $Q_{i}$ an agent has learned in previous interactions, the probability of opinion expression follows a softmax choice model of the form
(3)$$p_{i}=\frac{1}{1+e^{-\beta Q_{i}}}$$
in which β governs the rate of exploration. Taken together, the action selection (Equation 3) and the TDRL scheme (Equation 2) naturally account for the effect that agents become more (less) willing to speak out after receiving positive (negative) feedback.

### 2.3. Group Setting

Assume that we can characterise the two opinion groups $G_{1}$ and $G_{2}$ in terms of their sizes ($N_{1}$ and $N_{2}$), their in-group cohesion and intergroup connectivity. The probability of in-group influence is $q_{11}$ for group 1 and $q_{22}$ for group 2. The interaction probability across groups is denoted by $q_{12},q_{21}$, respectively. We assume that these interaction probabilities are equal for all agents within the same opinion group. Following a mean-field approach, we derive a dynamical system governing the average behaviour of agents in $G_{1}$ and $G_{2}$. That is, we are interested in the average values of opinion expression $Q_{1}=\frac{1}{N_{1}}\sum_{i\in G_{1}}Q_{i}$ and $Q_{2}=\frac{1}{N_{2}}\sum_{i\in G_{2}}Q_{i}$ and their evolution. For further details and a mathematical justification of this group-level description the reader is referred to [48].

Given the group sizes $N_{1}$ and $N_{2}$ and the homogeneous interaction probabilities $q_{11},q_{22},q_{12}$ and $q_{21}$, we define the *structural strength* of $G_{1}$ and $G_{2}$ (denoted as γ and δ) as
(4)$$\gamma =\frac{\left(N_{1}-1\right)}{N_{2}}\frac{q_{11}}{q_{12}}\;\mathrm{and}\;\delta =\frac{\left(N_{2}-1\right)}{N_{1}}\frac{q_{22}}{q_{21}}.$$
The structural strength of a group is determined by the relative size of the group and the relative in-group connectivity or *cohesion* [49,50]. As γ and δ determine the probability of in-group versus out-group interaction (γ/(γ+1) versus 1/(γ+1) for group 1), they also govern the expected rewards for opinion expression for the two groups with
(5)$$E\left(r_{1}\right)=p_{1}\frac{\gamma }{\gamma +1}-p_{2}\frac{1}{\gamma +1}-c,$$
for opinion group $G_{1}$ and
(6)$$E\left(r_{2}\right)=p_{2}\frac{\delta }{\delta +1}-p_{1}\frac{1}{\delta +1}-c,$$
for $G_{2}$. Note that the probabilities for opinion expression $p_{1}$ and $p_{2}$ are given by (Equation 3) substituting the agent index *i* by the respective group index. As an example, consider an agent in $G_{1}$ when expressing its opinion (Equation (Equation 5)). With a probability of $\frac{\gamma }{\gamma +1}$, the agent’s neighbour will be in $G_{1}$ as well and provide positive feedback with probability $p_{1}$. With probability $\frac{1}{\gamma +1}$, the agent will meet a neighbour in the opposing opinion group $G_{2}$ and receive negative feedback when agents in $G_{2}$ are expressive (i.e., with $p_{2}$).

As visible in Equation (Equation 2), in TD learning the change of Q-values from one time step to the other is given by the TD error times the learning rate α. Similarly, at the group level the update of the Q-values $Q_{1}$ and $Q_{2}$ from one time step to the next can be written as
(7)$$\begin{array}{c} \Delta Q_{1}=Q_{1}^{t+1}-Q_{1}^{t}=\alpha \left(E\left(r_{1}\right)-Q_{1}\right)\\ \Delta Q_{2}=Q_{2}^{t+1}-Q_{2}^{t}=\alpha \left(E\left(r_{2}\right)-Q_{2}\right) \end{array}$$
where we introduce the expected group rewards $E(r_{1})$ and $E(r_{2})$ derived above. In the continuous time limit [51,52,53], we replace t+1 by t+δt and α by αδt and take δt→0. This allows us to describe the model dynamics as a system of two differential equations
(8)$$\begin{array}{c} \dot{Q_{1}}=E\left(r_{1}\right)-Q_{1}\;\\ \dot{Q_{2}}=E\left(r_{2}\right)-Q_{2}, \end{array}$$
where we can omit the prefactor α by rescaling time. (When performing the continuous time limit, we have to rescale the learning rate $\alpha^{\prime }$ with $\alpha^{\prime }=\frac{\alpha }{\delta t}$. Thus, the equations would have the form ${\dot{Q}}_{g}=\alpha^{\prime }\left(E\left(r_{g}\right)-Q_{g}\right)$. Without losing any generality, we can set $\alpha^{\prime }=1$ by rescaling time.) As the right hand side is zero when the Q-value estimate is equal to the expected reward, the fixed points of (Equation 8) are possible equilibria of the associated collective game. (A game-theoretic analysis of the model was presented in [48]).

## 3. Applications

### 3.1. Organized Minorities and Silent Majorities

We applied this model to a minority–majority setting in which one third of the population supports opinion 1 and the other two thirds hold the majority view opinion 2. The group size ratio $\left(N_{2}-1\right)/N_{1}$ approached two for a large number of agents. In the first scenario, the interaction probabilities were homogeneous over the entire population ($q_{11}=q_{22}=q_{12}=q_{21}=q$). This corresponded to the Erdos–Rényi random graph [54,55] with link probability *q* and represented a situation without any particular organisation of social relations within and in between both camps. The structural strength indicators (Equation 4) were then determined by the relative group sizes: γ=1/2 and δ=2. For example, consider that there are $N_{1}=100$ agents in the minority and $N_{2}=200$ in the majority group, and let q=0.05 such that each agent has 15 links on average. In this unstructured case, agents from the minority are connected to 5 agents of the in-group and to 10 agents of the out-group, whereas an agent in the majority has an expected number of 10 neighbours in its own majority group and only 5 out-group connections. In this random graph setting, the dynamical system (Equation 8) has only one stable fixed point at $Q_{1}\approx -0.66$ and $Q_{2}\approx 0.66$. This situation is shown on the left-hand side in Figure 1, where the respective fixed point is marked by the blue circle. The associated expression rates are $p_{1}\approx 0.067$ for the minority and $p_{2}\approx 0.995$ for the majority. That is, the majority is expressive and the minority silent. The phase plot shows that even if expressive in the beginning (i.e., $Q_{1}>0$), agents in the minority find less and less support for their opinion and increasingly avoid expressing their opinion in public.

However, the minority can gain public impact if the internal organization of the group becomes more cohesive. The effect of this structural transition towards a stronger minority’s organisation is shown in Figure 1 and Figure 2. Figure 1 shows the phase portraits of Equation (Equation 8) for three different values of γ=1/2,2,4 that result from an increasing connectivity in the minority group ($q_{11}$). It also shows the respective isoclines of the dynamical system and the stable fixed points at their intersection. As the probability $q_{11}$ of in-group connections increases, the system undergoes a series of saddle-node bifurcations. First, a small increase of $q_{11}$ (and hence γ) gives rise to an additional stable fixed point in which only the minority is expressive (not shown in Figure 1, yellow regime of competition in Figure 2). The minority and majority compete for public voice. As the internal connectivity of the minority group increases to $q_{11}=4q$, the situation becomes symmetric with γ=δ=2. In other words, the minority can compensate its quantitative inferiority by a more cohesive internal organisation. Both groups can readily express their opinion if the other group is silent (competition, Figure 1, centre). However, an additional stable fixed point in which both opinions coexist also appears through another saddle-node bifurcation (coexistence). Finally, if the internal cohesion of the minority group becomes very large ($q_{11}=7q$), the fixed point associated to a loud majority and silent minority disappears. That is, the minority always voices their view in public while the majority may become silent (see Figure 1, r.h.s.).

### 3.2. “Spirals of Silence” as a Particular Regime of a More General Process

In the model, agents observe and react to their social environment in a way that is strongly reminiscent of Noelle-Neumann’s theory of the spiral of silence [15,16,18]. In repeated interaction within their local neighbourhood, agents form a “quasi-statistical” impression of the current opinion climate in terms of an internalized expectation (*Q*-values) of which opinion is prevalent in their public spheres and whether their opinion can be articulated without being sanctioned. If their opinion corresponds to the perceived majority view, they become more willing to speak out. If they perceive themselves to hold the minority view, they become less willing to do so. If all agents adapt to the current opinion landscape in this way, minorities are forced into a spiralling process in which silence leads to more silence. However, this happens only if the minority is perceived as minority in both groups. A bifurcation analysis of our model shows (see Figure 2) that majorities can also be forced into silence if a minority acts as a cohesive whole. Even a slight increase of homophily with respect to minority interaction can lead to a situation where a loud minority dominates public discourse because the majority is silent. Individuals with the actual majority opinion learn that voicing their view in public is rarely answered by support and is more often challenged by an expressive minority. The silence of the majority group is then collectively reinforced because each individual member is worse off by expressing their opinion.

The spiral of silence theory emerged as an attempt to explain a series of “last-minute swings” during German elections in the sixties and seventies [16]. (Termed “bandwagon effect”, this phenomenon had already been observed by Lazarsfeld and colleagues in the 1940 US presidential elections [56]). While surveyed voting intentions where head-to-head between the two major parties until the very last days of the campaigning period, the evolution of expectations about who would win the election showed a clear trend towards the final winner during the month before the election day. Developing a series of refined survey instruments, Noelle-Neumann showed that differences in the willingness to publicly support a party were one source of these trends. Our model captures this dynamical feedback between the internalized expectations of the majority and the willingness to actively speak out for one’s party and suggests that the situation of election campaigns at that time is characterised by the competitive regime in Figure 2 and Figure 3.

Our research shows that the assumptions underlying the spiral of silence theory are well-grounded in neuroscientific research on the processing of social feedback. However, our model shows that the collective process described by Noelle-Neumann—that is, the spiral of silence—is only one possible outcome of the individual-level assumptions on which the theory builds. The bifurcation analysis of the dynamical system (Equation 8) reveals that the structural transformations of group interaction may lead to qualitatively different regimes of collective opinion expression, including echo chambers.

### 3.3. Social Feedback and Echo Chambers

Today, social media are rapidly transforming the landscape of public opinion expression providing niches for virtually every opinion. Social network services have flexibilized options to connect with like-minded others no matter the topic or political stance—may that be on social media under hashtags such as #MeToo, on Telegram channels [57] or on imageboards with extremist content [58]. These fragmented online spheres hence provide previously unseen opportunities to escape the “fear of isolation” and to learn that there are others who share a similar view.

In the model, group interaction that is more and more structured around shared opinions is captured by a simultaneous increase of γ and δ meaning that social interaction with like-minded agents becomes more probable for both opinion groups. The qualitative effects of this structural transition towards more assortative networks is shown in Figure 3. As in-group ties become more prevalent in both groups, the system undergoes two saddle-node bifurcations from a competitive regime where only one group is aloud to a regime where coexistence is the only stable outcome. Private or semipublic rooms for expressing opinion online act as “echo chambers” and enable opinions previously marginalized or placed under taboo to resist the spiral of silence and become salient in the more general public discourse.

Our model entails that the perception of public support for an opinion is biased not only by the local connectivity of individuals [59], but also by the willingness of the supporters of different opinions to engage in the media. If opinions are shaped in social circles sorted around opinion, opposing opinion groups find their own views backed with social support and, in turn, become convinced of their primacy. The effect of ideological asymmetries [60,61] in opinion expression might overcome local homophily biases in the social network structure, since neutral observers and lurkers also get confronted with this distorted impression of public opinion. Democratic societies currently struggle with this transformed multifaceted climate of opinions because the foundational idea of government built on the common ground of public opinion [62,63] is fundamentally challenged.

## 4. Discussion

The SFT aims to contribute to a better understanding of societal-level implications of human social nature in modern information society. It provides a link between recent research on the neurological basis of social behaviour and the sociological theory of public opinion formation and expression. The model presented in this paper involved abstractions and assumptions at three different analytical levels (see Figure 4) each being subject to intensive research from different disciplinary angles. Providing a coherent theoretical account that integrates sociological modes of structural explanation [3,5], adaptive decision theory [28,44] and its underlying neurological mechanisms [64,65], the SFT offers a unique framework for guiding future interdisciplinary research on how social and cognitive mechanisms involved in platform-mediated communication on opinions play out at the scale of larger collectives.

At the collective level (top row), the SFT relates structural transformations in how we interact with one another to different regimes of collective opinion expression. The main modelling assumption made at this level is to map complex networks of social interaction to the relations within and across groups. Network science has brought about a portfolio of graph models to more realistically capture social interaction patterns [1,49,59,66], to which the model but not necessarily its formulation as a 2D system of differential equations can be applied. On the other hand, our theory suggests that empirical networks inferred on the basis of digital trace data [67] may be inherently biased by the activity of users who learnt that interaction on the media is rewarding. In fact, our model suggests that retrieved interaction patterns such as retweet networks [68,69,70] may render a situation more polarized than it actually is, because public expression is less rewarding for actors who maintain relations across different opinion camps. Research on Twitter has also shown that retweets and replies give rise to very different global patterns of group interaction [61] suggesting that they serve rather different communicative functions. By bridging from individual decisions to express opinions to emergent collective activity patterns, the SFT provides a useful theoretical framework to analyse how those different communicative functions play out at large.

In the model, the micro level of social interaction (second row) is conceived as repeated opinion expression games in which agents respond to one another with signals of approval or disapproval. This entails simplifications such as dyadic interaction and a reward system that is homogeneous across individuals and groups. However, by drawing on games for modelling individual interaction, the SFT is well-equipped to take into account individual differences in reward perception as well as characteristics of the incentive structure of different social media platforms. In contrast to most previous models of social learning and opinion dynamics, the SFT takes into account that users have to express their opinions within the technical constraints of a platform in question. Conceiving social interaction as communication games that account for the incentives to engage online shifts the explanatory focus from forms of social influence to the rewards and incentives of opinion expression in different online settings. Of note, opinion games are also flexible enough to include cognitive costs associated to, for instance, preference falsification [7,71] and other sources of cognitive dissonance [72].

The social feedback framework draws on a neurocognitive foundation of TDRL (bottom row). In order to demonstrate that biologically rooted mechanisms of reward and value processing capture collective processes described in, for instance, the spiral of silence theory, we relied on the most simple TDRL scheme in the model. Social neuroscience is quickly advancing towards a better understanding of how brain areas related to cognitive control interfere with this basic reward circuit. Recent work has revealed, for instance, that neural responses to social feedback are influenced by the social relation with the interaction partner [73] and that the reward valuation circuit is highly involved in shaping these relations [74]. Experimental designs that mimic interaction on social media [29,30] could clarify the role of different incentive systems for online engagement on opinion. This would contribute to a more systematic understanding of the types of games that are played in social media environments.

## 5. Conclusions

Massive social interaction in modern information society favours fast and largely unconscious modes of information processing. The model developed in this paper showed that the basal brain processes governing our reactions to social approval and disapproval could have a tremendous impact on collective processes of opinion expression. Simple feedback mechanisms may be at the root of phenomena such as silent majorities and enable well-organized minority groups to gain public voice or even dominate a discourse. Social media that facilitate massive and strategic social organisation around opinions can fundamentally alter the perception of public opinion in a society.

Social feedback theory has been proposed as a modelling framework that more explicitly takes into account the decision processes involved in expressing opinions online [17]. It extends previous work in opinion dynamics by allowing agents to refrain from participating in opinion exchange processes under certain circumstances and in certain environments. This has practical implications for computational social science methods aiming to measure opinions on online data where only users that actively engage in opinion exchanges become visible. However, for the sake of mathematical tractability, the current model was limited in terms of social interaction networks and did not account for different means of communication that online platforms may provide. Future models have to more realistically map the distinctive features and affordances of real social media platforms to become a practical tool for exploring digital communication devices that better serve deliberative modes of online opinion exchange.

## Figures and Tables

**Figure 1 entropy-24-01484-f001:**
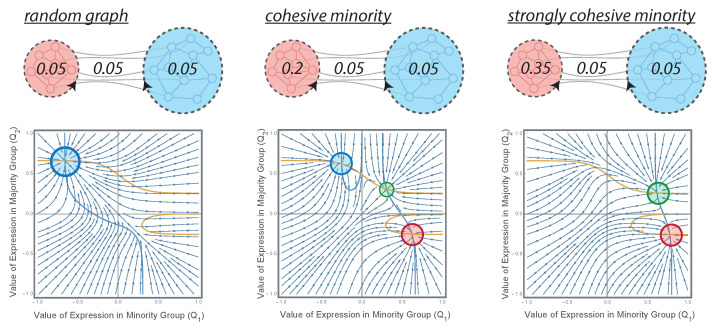
Two groups supporting two different opinions struggle for public expression. The majority group (blue) is twice as big as the minority group (red). The three different situations represent subsequent increases of internal cohesion of the minority group and their effect on the respective phase dynamics of the system. The phase portraits show the evolution of $Q_{1}$ and $Q_{2}$ towards the fixed points of Equation (Equation 8). The isoclines of the dynamical system are shown and the stable fixed points at their intersection are coloured according to whether the majority (blue circle), the minority (red circle), or both (green circle) are expressive. While minority expression is unstable in an unstructured random graph, the minority can compensate their quantitative inferiority by a stronger internal organisation. Results for an exploration rate β=8 and c=0. Random graph (**left**): ER graph with link probability $q_{11}=q_{22}=q_{12}=q_{21}=0.05$. For group sizes $N_{1}=100$ and $N_{2}=200$, each agent has 15 links on average. The minority is connected to 5 agents of the in-group and to 10 agents of the out-group and vice versa for the majority leading to structural strength indicators γ=0.5 and δ=2. The resulting system has only one stable fixed point at $Q_{1}\approx -0.66$ and $Q_{2}\approx 0.66$ with associated expression rates of $p_{1}\approx 0.067$ and $p_{2}\approx 0.995$. That is, only the majority group is expressive and the minority silent. Cohesive minority (**centre**): Increasing internal organisation of the minority group by raising the connection probability within the minority to $q_{11}=0.2$. This increased group cohesion is reflected in an increased structural strength γ=2. δ is not affected. The system becomes symmetric and minority (red circle) or majority group expression (blue circle) are solutions reached depending on the initial values of expression. An additional fixed point (green circle) emerges in which two groups are loud. Strongly cohesive minority (**right**): The in-group cohesion of the minority further increases ($q_{11}=0.35$) leading to γ=4 and δ=2. The case that only the majority is in expression mode is no longer stable and the minority will always express its opinion. Coexistence is still possible.

**Figure 2 entropy-24-01484-f002:**
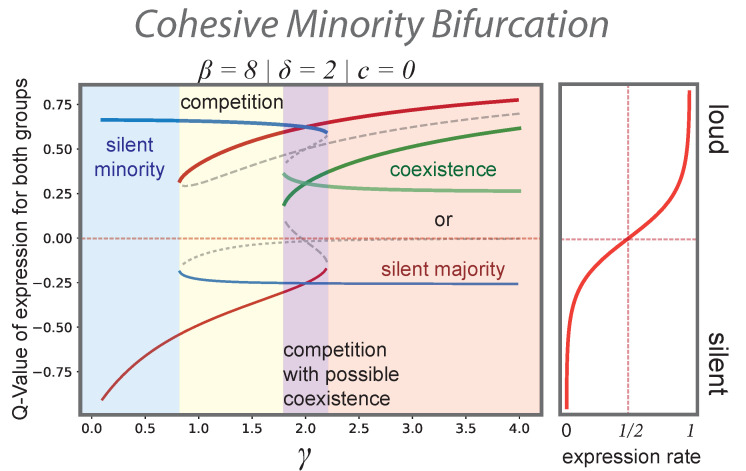
Bifurcation plot of the scenario in which a minority ($N_{1}/N_{2}=1/2$) gains public voice through stronger internal organisation (see also Figure 1). On the right-hand side, the expression rate (Equation 3) is shown as a function of *Q* for β=8. The strength δ of the second group is kept constant (δ=2) and the costs of expression are zero (c=0). As the internal cohesion of the minority group increases, for instance, due to strategic linking or tying group symbols, the system undergoes a series of saddle-node bifurcations. The minority’s expression becomes more and more likely. Solid lines show the Q-values at the stable fixed points. For equilibria in which only one group is loud, the blue lines represent the majority group $Q_{2}$, the red lines the minority $Q_{1}$. The green lines correspond to the coexistence equilibrium in which both groups are expressive. While an unorganised minority is forced into silence (blue regime), a slight increase in group cohesion makes the minority’s expression a stable outcome if the majority is silent (competition, yellow). At a certain point ($q_{11}=4q$ and γ=δ=2), the minority’s organisation can compensate numerical inferiority and a stable coexistence of two expressive groups is possible. By further increasing the minority group’s strength, it is always visible in public while the majority may enter a spiral of silence.

**Figure 3 entropy-24-01484-f003:**
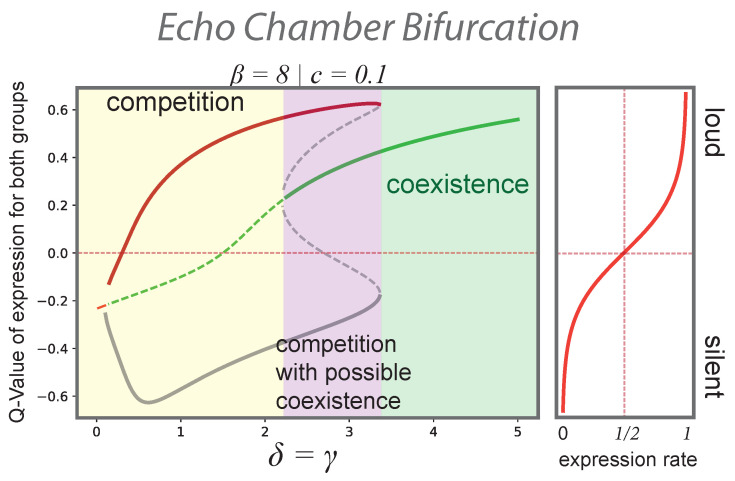
Bifurcation plot of the scenario in which two groups of equal size become more structured around the opinion they support. An increase of homophily for both opinion groups is captured by increasing γ and δ at the same time. The situation is symmetric and only $Q_{1}$ is shown. After a phase of competition, if homophily is low (yellow), coexistence emerges as a fixed point (violet) and becomes the only solution after a further slight increase of homophily (green). Both groups express their opinion within their own niches. (Results for β=8 and c=0.1).

**Figure 4 entropy-24-01484-f004:**
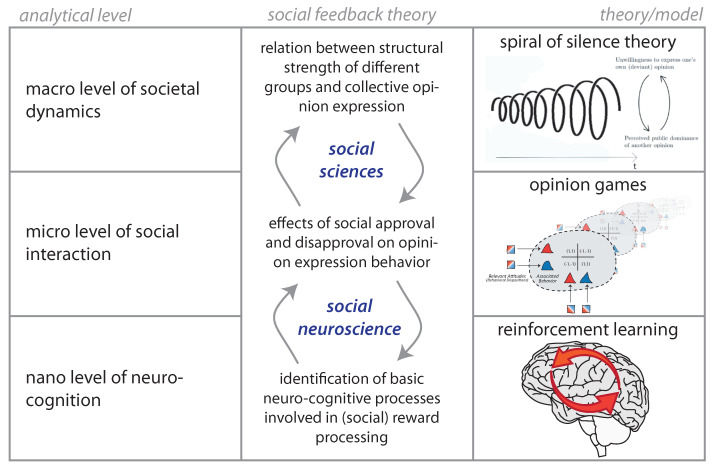
Schematic summary of social feedback theory of opinion expression. The theory involves three analytical levels from the level of neurocognitive processes to the level of social interaction, to the macro level of collective dynamics. The SFT bridges these levels through the notion of opinion games: first, by assuming that agent behaviour and the associated expected rewards adapt according to a reinforcement learning scheme that accurately models the reward processing system in the brain; and second, by bridging from individual decisions to express opinions to emergent patterns of collective opinion expression.

## Data Availability

Not applicable.

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
