# Peer review of "Modelling Spirals of Silence and Echo Chambers by Learning from the Feedback of Others"

_entropy, 2022, doi:10.3390/e24101484_

Round 1

Reviewer 1 Report

The research "Modelling spirals of silence and echo chambers by learning from the feedback of others", present the role that basic mechanisms of social information processing play in massive computer mediated interaction on opinions.

Congratulations to the authors, it is very well written and the methodology correctly developed and applied, just a few minor comments:

 - In the conclusions, the authors must add a paragraph on limitations of the research, future work and practical implications of the research.

- In section 3. Applications, the authors indicate that: "We apply this model to a minority-majority setting in which one third of the population supports opinion 1 and the other two thirds hold the majority view opinion 2. (line 112)". It is not clear the size of the selected sample, if it existed, or the amount of simulated data used to carry out the application, or indicate the building of the dataset, the authors should clarify the above.

Author Response

Thank you for this positive and encouraging feedback on our manuscript. We have addressed the two points as follows:

Reviewer:

- In the conclusions, the authors must add a paragraph on limitations of the research, future work and practical implications of the research.

Response:

Thanks. We added one paragraph to the conclusion trying to reflect how to overcome the obvious limitations of this simple model and work towards models that are of more practical use:

„Social feedback theory has been proposed as a modeling framework that more explicitly takes into account the decision processes involved in expressing opinions online \citep{Banisch2021}. It extends previous work in opinion dynamics by allowing that agents might refrain from participating in opinion exchange processes under certain circumstances and in certain environments. This has practical implications for computational social science methods aiming to measure opinions on online data where only users that actively engage in opinion exchange become visible. However, for the sake of mathematical tractability the current model is limited in terms of social interaction networks and does not account for different means of communication that online platforms may provide. Future models have to more realistically map the distinctive features and affordances of real social media platforms so to become a practical tool for exploring digital communication devices that better serve deliberative modes of online opinion exchange.“

Reviewer:

- In section 3. Applications, the authors indicate that: "We apply this model to a minority-majority setting in which one third of the population supports opinion 1 and the other two thirds hold the majority view opinion 2. (line 112)". It is not clear the size of the selected sample, if it existed, or the amount of simulated data used to carry out the application, or indicate the building of the dataset, the authors should clarify the above.

Response:

All the analysis is based on the group formulation as a dynamical system (Eq. 7). To provide better intuition we exemplify the meaning of the parameters by adding (beginning of Sec. 3.1):

„For example, consider that there are $N_1 = 100$ agents in the minority and $N_2 = 200$ in the majority group, and let $q = 0.05$ such that each agent has 15 links on average. In this unstructured case, agents from the minority are connected to 5 agents of the in-group and to 10 agents of the out-group whereas an agent in the majority has an expected number of 10 neighbors in its own majority group and only 5 out-group connections.“

Reviewer 2 Report

This is a quality paper with a nice contribution to a timely topic. The proposed model captures the mechanism by which the majority is silenced and minority groups become public if they are well organized. It is also interesting that the proposed model can be used to explain the origin of echo-chompers. Although the proposed model and the analysis is rather too simple, but I would like to recommend the acceptance because it is a short paper but a nice one.

Author Response

Dear Referee 2,

thank you for this positive and encouraging feedback on our draft.